# Ultrastructural Evaluation of Mouse Oocytes Exposed In Vitro to Different Concentrations of the Fungicide Mancozeb

**DOI:** 10.3390/biology12050698

**Published:** 2023-05-10

**Authors:** Marta Gatti, Manuel Belli, Mariacarla De Rubeis, Mohammad Ali Khalili, Giuseppe Familiari, Stefania Annarita Nottola, Guido Macchiarelli, Edmond Hajderi, Maria Grazia Palmerini

**Affiliations:** 1Department of Anatomy, Histology, Forensic Medicine and Orthopaedics, Sapienza University, 00161 Rome, Italy; 2MEBIC Consortium, IRCCS San Raffaele Roma, 00166 Rome, Italy; 3Department of Life, Health and Environmental Sciences, University of L’Aquila, 67100 L’Aquila, Italy; 4Department of Reproductive Biology, Yazd Reproductive Sciences Institute, Shahid Sadoughi University of Medical Sciences, Yazd 8916877391, Iran; 5Department of Pharmaceutical Sciences, Catholic University Our Lady of Good Counsel, 1000 Tirana, Albania

**Keywords:** oocytes, mancozeb, ultrastructure, transmission electron microscopy, fertility, reproductive toxicity

## Abstract

**Simple Summary:**

Exposure to endocrine-disrupting pollutants, such as the fungicide mancozeb, is linked to various environmental health hazards, including female fertility. Although the dithiocarbamate mancozeb has low reported toxicity in mammals, it impairs female reproductive functions in exposed animals and humans. The specific mechanism of action of mancozeb and the damage to cell structures in the female reproductive system are still unclear. This study aims to describe the ultrastructure of mouse oocytes exposed in vitro to increasing concentrations of mancozeb (0.001–1 μg/mL) by light and transmission electron microscopy and to perform a morphometric analysis over significant organelles. While from 0.001 to 0.1 μg/mL, oocyte ultrastructure was comparable to controls, at the highest concentration (1 µg/mL), a decrease in the numerical density of mitochondria and cortical granules, an altered organelle distribution, and flattening of microvilli were observed. These results could be responsible for the adverse effect of this fungicide on mammalian reproductive performance.

**Abstract:**

Mancozeb is a widely used fungicide, considered to be an endocrine disruptor. In vivo and in vitro studies evidenced its reproductive toxicity on mouse oocytes by altering spindle morphology, impairing oocyte maturation, fertilization, and embryo implantation. Mancozeb also induces dose-dependent toxicity on the ultrastructure of mouse granulosa cells, including chromatin condensation, membrane blebbing, and vacuolization. We evaluated the effects on the ultrastructure of mouse oocytes isolated from cumulus-oocyte complexes (COCs), exposed in vitro to increasing concentrations of mancozeb. COCs were matured in vitro with or without (control) low fungicide concentrations (0.001–1 μg/mL). All mature oocytes were collected and prepared for light and transmission electron microscopy. Results showed a preserved ultrastructure at the lowest doses (0.001–0.01 μg/mL), with evident clusters of round-to-ovoid mitochondria, visible electron-dense round cortical granules, and thin microvilli. Mancozeb concentration of 1 μg/mL affected organelle density concerning controls, with a reduction of mitochondria, appearing moderately vacuolated, cortical granules, and microvilli, short and less abundant. In summary, ultrastructural data revealed changes mainly at the highest concentration of mancozeb on mouse oocytes. This could be responsible for the previously described impaired capability in oocyte maturation, fertilization, and embryo implantation, demonstrating its impact on the reproductive health and fertility.

## 1. Introduction

Over the past decades, due to the growing use of pesticides, the impact of environmental pollutants on human health represents a worldwide concern. Exposure to these substances may occur occupationally during agricultural and industrial activities through soil, air inhalation, contaminated food and water ingestion, and skin absorption [1,2,3,4]. These pesticides affect health both in animals and humans, as demonstrated by increased carcinogenesis [5,6], toxic effects on neuronal and immune systems [7,8], reproductive toxicity, and reduced fertility [9,10,11]. The duration and timing of exposure play a significant role in the severity of organ dysfunctions and metabolism-associated disorders [12].

Mancozeb, an ethylene-bis-dithiocarbamate, is a fungicide currently used to manage fungal diseases in plants. Introduced in the global market in 1962, it has been used for over 70 years due to its low price [13]. In 2021, EFSA (European Food Safety Agency) banned the use of mancozeb as a pesticide in Europe, due to its reproductive toxicity and endocrine-disrupting properties [14]. Nevertheless, in several countries, this fungicide is still largely employed. 

Mancozeb toxicity is widely attributed to the adverse effects of its metabolite, Ethylenethiourea (ETU), used to evaluate its exposure in humans [15]. In workers exposed to 147.11 µg concentration of fungicide for 38 workdays, the ETU concentration in the urine ranged from 0.8 to 61.4 μg/L [16,17,18]. High doses of mancozeb metabolites (from ng/L to mg/L) are often detected in the soil and surrounding water [19]. Mancozeb was detected at a concentration of 39 μg/L in environmental water near farms [20]. The US. EPA (Environmental Protection Agency) (2013) reported a concentration of mancozeb in surface water ranging from 0.1 to 25.2 µg/L [21]. In the air of California, values of mancozeb were comprised between 0.29 µg/m^3^ to 1.81 µg/m^3^ [22]. Despite its short environmental resistance, ETU persists longer in the soil (5–10 weeks). In 1988, the WHO (World Health Organization) estimated exposure to mancozeb and ETU was 0.01–1 µg/kg b.w./day for the general population. Small quantities of ETU were detected in tobacco, fruits, and vegetables. Agricultural and farming activities, during field application, represent the main sources of contamination for the general population through inhalation, direct skin contact, and food intake [23]. 

Mancozeb and its metabolite product are considered by IARC (International Agency for Research on Cancer) Class 3 carcinogen, given the limited evidence in humans, although teratogenic and carcinogenic effects have been observed in animal studies [24]. In vitro experiments evidenced neurodevelopmental damages after mancozeb and ETU exposure, due to mitochondrial respiration inhibition (2.6–31.2 µg/mL), active ROS generation (0.08–7.8 µg/mL) [7,25]; oxidative stress induction, genotoxicity, and apoptosis were revealed in rat fibroblast (0.125–0.5 µg/mL) [26].

Endocrine disrupting chemicals (EDCs) such as mancozeb are defined as “exogenous agents that can potentially mimic, secrete, carry natural hormones, or replace them by interacting with their receptors” [27]. Even though the involvement of EDCs, such as mancozeb, in certain reproductive diseases is well documented [28], only a few studies have examined the direct impact of pesticides on fertility, especially in female infertility. It has been hypothesized that the negative effect of these toxicants on reproductive health occurs either by affecting the hypothalamic-pituitary-gonadal axis or by directly affecting the genital organs through cytotoxicity on germ cells [29,30]. 

Despite low acute mammalian toxicity (LD_50_ = 8 g/kg/day in rats), long-term and chronic exposure to mancozeb could lead to spontaneous miscarriage, maternal mortality, and fetal malformation in rats and rabbits [31]. It affected pregnancy and embryo development by apoptosis and induced gonadal toxicity in female rats by genotoxic and malignant alterations in human ovarian cells [32,33]. Mancozeb impaired the endometrium receptivity through the direct suppression of prostaglandin E synthase (PGS) expression in the uterine microenvironment [34] and affected mouse embryo implantation by reducing trophoblastic spheroids (embryo surrogates) attachment onto endometrial epithelial cells via downregulation of estrogen receptor β (ERβ) and integrin β3 (ITGβ3) expression [35]. 

Nevertheless, this fungicide’s exact mechanism of action on reproductive functions still needs to be fully understood. Exposure to a mancozeb concentration of 0.003 µg/mL, impairs mouse embryo development in vitro by inducing blastomere apoptosis [36]. As recently described, exposure to mancozeb (0.3–30 µm/mL) reduces cumulus cell expansion, indicating insufficient maturation of the caprine oocyte cytoplasm upon high-dose exposure [37]. The cytotoxic effects of mancozeb on the caprine oocytes were seen at the highest concentrations (3 and 30 μg/mL), with a significant decrease in the nuclear maturation rate by preventing the formation of the metaphase plate. Interestingly, disintegrated nuclear materials formation was present even at low concentrations (0.3 μg/mL) [37]. Transcriptomic analysis on ovaries from in vivo exposed mice (100 mg/kg) revealed an abnormal mitochondrial respiratory chain function responsible for oxidative phosphorylation decoupling, oxidative stress, ovarian injury, and apoptosis [38].

By using a lower range of concentrations (0.001–1 µg/mL), we previously observed alterations of the spindle morphology in mouse cumulus-oocyte complexes (COCs) associated with a reduction of the fertilization rate in vitro [39]. A reduced ability to sustain the early steps of fertilization, i.e., two pronuclei (2PN) formation, was also confirmed in vivo [40]. Numerous studies in mouse granulosa cells (GCs) highlighted p53 downregulation, ROS production increase, and mitochondrial activity alterations after fungicide exposure [32,41]. All these data were further sustained by ultrastructural analysis of mouse GCs, evidencing chromatin condensation, membrane blebbing, cytoplasmic vacuolization, and cell degeneration [42]. Damages to the somatic compartment of COCs may be co-responsible for the above-cited germ cell alterations, given the close association between GCs and the oocyte. 

However, to better clarify the potentially detrimental effects of mancozeb exposure to mouse oocytes, we performed a morphological study by light (LM) and transmission electron microscopy (TEM) and a morphometric evaluation of cytoplasmic organelles.

## 2. Materials and Methods

### 2.1. Chemicals

Unless otherwise stated, all materials were purchased from Sigma Chemical (St. Louis, MO, USA). 

### 2.2. Animals

Swiss CD1 mice (Harlan Italy, Udine, Italy) were housed in individual cages with a 12:12 h light: dark cycle, controlled temperature (21 ± 1 °C), and free access to food and water. Twelve prepubertal females (21–23 days old) were intraperitoneally administered 5 IU of PMSG (Pregnant Mare Serum Gonadotropin) (Intervet, Milan, Italy) and euthanized 48 h later [42]. Animals were maintained according to the Italian Department of Health Guide for Care and Use of Laboratory Animals [42].

### 2.3. In Vitro Maturation of Oocyte-Cumulus Cell Complexes and Experimental Protocol

COCs were recovered by puncturing antral follicles from mouse ovaries with an insulin syringe. COCs (10/group) were matured in vitro (IVM) at 37 °C and 5% CO_2_ in air in alpha MEM supplemented with 0.23 mM pyruvate and two mM l-glutamine, with or without (control) low concentrations of the fungicide ranging from 0.001 to 1 μg/mL. 

These concentrations were selected based on the results of previous studies on mouse oocytes and granulosa cells treated with mancozeb [40,42]. A stock solution of mancozeb (50 µg/mL, 50×) was prepared by resuspending the fungicide in the culture medium or vehicle (DMSO: dimethyl sulfoxide) to obtain the desired concentrations by diluting with the culture medium. The final volume of vehicles added to the samples never exceeded 0.1% (*v/v*), and no adverse effects on maturation were observed [39]. 

After 16 h, control and mancozeb-exposed oocytes were deprived of cumulus cells and arrested at metaphase II (MII), as evidenced by extrusion of the first polar body (PB1). Experiments were repeated in triplicate.

### 2.4. Light Microscopy (LM) and Transmission Electron Microscopy (TEM)

After collection, oocytes were immediately fixed in 2.5% glutaraldehyde (Agar Scientific, Cambridge Road Stansted Essex, Cambridge, UK) in PBS (phosphate buffered saline, pH = 7–7.4) and were maintained at 4 °C until the next preparative for TEM observations [42,43,44,45,46]. After several washes in PBS, mouse oocytes were post-fixed with 1% osmium tetroxide (electron microscopy sciences) in PBS for one and half hours in a dark compartment at 4 °C. Oocytes were then embedded in small blocks of 1% agar of about 5 mm × 5 mm × 1 mm in size, dehydrated in an ascending series of ethanol, immersed in propylene oxide for solvent substitution, and left overnight in a propylene oxide/resin 1:1 solution. Finally, they were embedded in epoxy resin EMbed-812 (Electron Microscopy Sciences, 1560 Industry Road, Hatfield, PA, USA) and sectioned using an Ultracut E ultramicrotome (Leica EMUC6, Wetzlar, Germany). Semithin sections (1 μm thick) were stained with methylene blue (Sigma-Aldrich), examined by LM (Zeiss Axioskop), and photographed using a digital camera (Leica DFC230). 

Ultrathin sections (70–90 nm thick) were cut with a diamond knife, mounted on copper grids, and contrasted with Uranyless (Uranyl acetate alternative) (TAAB Laboratories Equipment Ltd., Aldermaston, UK) and Lead Citrate (Electron Microscopy Science) before being examined and photographed using Zeiss EM10 and Philips TEM CM100 electron microscopes operating at 80 kV.

The following parameters were evaluated by LM and TEM and taken into consideration for the qualitative morphological assessment of the ultrastructural characteristics of oocytes: general features (including shape and dimension); cytoplasmic organization and cell organelles (state of preservation, type, distribution); oolemma (membrane integrity, presence, appearance, and distribution of microvilli); perivitelline space (width, presence of cytoplasmic fragments) and zona pellucida.

### 2.5. Morphometric Analysis

ImageJ 1.54 v software was used to measure the numerical density of mitochondria, SER, autophagic vesicles, multivesicular bodies and dense lamellar bodies, cortical granules, and microvilli, following observations of low-magnification TEM micrographs of control and mancozeb-treated oocytes [47,48]. More specifically, the numerical density of mitochondria, vesicular and tubular SER elements, autophagic vesicles, multivesicular bodies, and dense lamellar bodies was determined on at least five equatorial sections (distance between the sections: 3 μm) of two oocytes/group. Values are expressed as numerical density per 50 μm^2^ of the oocyte area. The evaluation of cortical granule and microvilli density was performed by analyzing TEM micrographs at 2500× of the entire surface profiles on five equatorial sections [49] of two oocytes/groups. Values are expressed as cortical granule and microvilli density (n. of cortical granules and microvilli per 10 micrometers of the oocyte linear surface profile). 

### 2.6. Statistical Analysis

All data are expressed as means ± standard deviation (SD). Statistical comparisons were performed using one-way ANOVA with Tukey’s honest significant difference (HSD) tests for post-hoc analysis (GraphPad InStat. GraphPad Software, La Jolla, San Diego, CA, USA). Differences in values were considered significant if *p* < 0.05.

## 3. Results

### 3.1. Controls

By LM on semithin sections, control oocytes showed a regular round shape with an equatorial diameter of about 80 µm. The ooplasm appeared dense and homogeneous, surrounded by an intact zona pellucida (Figure 1A, inset). 

In the ooplasm, TEM analysis revealed numerous organelles evenly distributed, indicating a high cytoplasm/organelle ratio (Figure 1A). Numerous round-to-ovoid-shaped mitochondria were mostly organized in groups throughout the ooplasm. Mitochondrial cristae were well-distinguishable, with a dense mitochondrial matrix; both inner and outer mitochondrial membranes appeared electron-dense (Figure 1B,D). In addition, a highly dense “cytoplasmic lattice,” a typical fibrillar structure, was diffusely scattered in the ooplasm of mouse oocytes. Lipid droplets and smooth endoplasmic reticulum (SER) were commonly found (Figure 1B). Numerous dense lamellar bodies were present, whereas multivesicular aggregates of small vesicles and large vacuoles were mostly located in the cortical region of the ooplasm (Figure 1C). Occasionally, immature autophagosome vesicles were dispersed throughout the cytoplasm. Numerous rounds and uniformly electron-dense cortical granules were linearly arranged below the oolemma (Figure 1B). A narrow perivitelline space was occupied by numerous, longer, and thin microvilli protruding from the oolemma. The zona pellucida appeared continuous and characterized by a smooth appearance (Figure 1B,C). Table 1 shows a summary of the main results.

### 3.2. Mancozeb 0.001 µg/mL

Oocytes exposed in vitro to the lowest concentration of mancozeb presented morphological features similar to the control group by LM. The zona pellucida was intact, and the ooplasm showed a high organelle density (Figure 2A, inset). 

Low magnification TEM micrographs confirmed a cytoplasmic ultrastructure similar to controls, with a homogenous distribution of numerous organelles in the ooplasm (Figure 2A); the cytoplasmic lattice was well-distinguishable and evenly distributed throughout the ooplasm. Mitochondria were the most prominent organelles. At higher magnification, the clustered organization of mitochondria and their ultrastructural characteristics, i.e., a round-to-ovoid shape, a double-layered electron-dense mitochondrial membrane, and a homogeneous matrix, did not change if compared to controls (Figure 2B,D). Rarely vacuolated mitochondria were present in the ooplasm. Numerous multivesicular bodies and SER were isolated or arranged in groups; membrane-bound lamellar bodies, containing a dense undigestible material were frequently detected in the ooplasm (Figure 2B,C). Occasionally, immature early autophagic vesicles with a marked lumen bordered by a double membrane were found (Figure 2D). Numerous rounded cortical granules with dark electron-dense content were located just beneath the oolemma (Figure 2C). Thick and short microvilli were regularly distributed along the oolemma, and the perivitelline space appeared narrow (Figure 2A,C). Table 1 shows a summary of the main results.

### 3.3. Mancozeb 0.01 µg/mL

By LM, oocytes treated with a concentration of 0.01 µg/mL of mancozeb had a round shape and an intact zona pellucida. The perivitelline space appeared less thick than in controls and mancozeb 0.001 µg/mL (Figure 3A, inset). 

Differently from the previous groups, TEM observations highlighted a reduction in organelle density and a patchy distribution (Figure 3A,B). The ooplasm was rich in clusters of round-shaped mitochondria with electron-dense inner and outer membranes. Occasionally, vacuolated mitochondria were detected in which the outer membrane and cristae were not distinguishable (Figure 3B–E). SER vesicles were visible (Figure 3B), together with multivesicular aggregates and multivesicular bodies. Fibrillar structures, typical of the cytoplasmic lattice, were distinctly observable (Figure 3B,C). There were numerous dense lamellar bodies and mature vesicles associated with autophagy (Figure 3C, inset). Subcortical round and electron-dense cortical granules were less abundant than in previous groups (Figure 3D). Microvilli appeared slightly shorter in length, lesser in number, and more irregularly distributed; moreover, extracellular material and debris were found in the perivitelline space (Figure 3C). The zona pellucida showed high density (Figure 3C). Table 1 shows a summary of the main results.

### 3.4. Mancozeb 0.1 µg/mL

Oocytes exposed to mancozeb 0.1 µg/mL showed a round-to-ovoid shape with a very narrow perivitelline space and an intact zona pellucida under LM (Figure 4A, inset). 

Ultrastructural analysis evidenced a dramatic reduction in organelle density (Figure 4A). Mitochondria appeared less numerous, with less evident mitochondrial cristae, compared to the previous group (0.01 µg/mL) (Figure 4A–C). Some of these organelles appeared vacuolated (Figure 4D). However, the quantity of vacuolized mitochondria remains almost unchanged compared to the previous group. Tubular SER elements seemed slightly reduced, compared to the previous group (Figure 4B). Multivesicular aggregates were distributed in the cortical region in the cytoplasm, accompanied by dense lamellar bodies (Figure 4B). Furthermore, TEM observations showed the presence of structures compatible with mature autophagic vesicles, enclosed by a single membrane and containing membranous material of unrecognizable origin (Figure 4C, inset). The density of cortical granules appeared to be reduced, whereas the oolemma showed smaller and shorter microvilli (Figure 4B). In the perivitelline space, it was detected the presence of extracellular material and debris (Figure 4C). The zona pellucida appeared thin and dense. Table 1 shows a summary of the main results.

### 3.5. Mancozeb 1 µg/mL

At the highest concentrations of mancozeb, oocytes showed signs of ultrastructural damage. By LM, the oocytes presented a round shape, and the ooplasm showed organelles condensed in one pole (Figure 5A, inset). 

TEM revealed a low cytoplasm/organelles ratio and a non-homogeneous distribution of organelles, which appeared reduced in some areas of the ooplasm compared to the previous groups (Figure 5A). Mitochondria were less numerous, with a round-to-ovoid shape, sometimes vacuolated. Outer mitochondrial membranes appeared highly electron-dense (Figure 5B). Clusters of mitochondria, occasionally including SER tubules, were interspersed with numerous isolated elements (Figure 5B, inset). Cytoplasmic lattices were identifiable in the ooplasm by fibrillar structures; multivesicular bodies and dense lamellar bodies were seen (Figure 5C,D). Noteworthy, some regions just beneath the oolemma, showed organelles-free areas and cortical granules were very rare (Figure 5A,B). Microvilli appeared mostly short, tiny, flattened, and not numerous, even if, in some areas they were longer. Sometimes, in the perivitelline space, extracellular materials, such as extracellular vesicles and cell fragments, were found (Figure 5B,D). The zona pellucida appeared thin and dense (Figure 5C,D). Table 1 shows a summary of the main results.

### 3.6. Morphometric Analysis 

The morphometric analysis revealed a downward trend in the mitochondrial numerical density from control to increasing concentrations of mancozeb (C: 28 ± 7.131; 0.001 µg/mL: 24.13 ± 5.436; 0.01 µg/mL: 23.38 ± 8.684; 0.1 µg/mL: 21.13 ± 4.794; 1 µg/mL: 18.25 ± 5.175). However, a significant decrease was evidenced between the group exposed to mancozeb 1 µg/mL and controls (18.25 ± 5.175 vs. 28 ± 7.131; *p* < 0.05) (Table 2). The density of multivesicular bodies, dense lamellar bodies, and vesicular/tubular SER elements did not reveal significant differences between groups (*p* > 0.05, Table 2). However, autophagic vesicles showed an upward trend at increasing mancozeb concentrations, even if not significant. 

The morphometric evaluation of cortical granules showed that control and mancozeb 0.001 µg/mL had a significantly lower number per 10 µm when compared to mancozeb 1 µg/mL (3 ± 0.7 and 3.2 ± 0.8 vs. 0.8 ± 0.8, respectively; *p* < 0.05), with a general declining trend, as seen for mitochondria (Table 2). The number of microvilli per 10 µm decreased at increasing concentrations of the pesticide, when compared to controls (15.6 ± 2.408), with a highly significant difference at 1 µg/mL (6 ± 2.449; *p* < 0.001) (Table 2). 

## 4. Discussion

This study described the effects of increasing concentrations of mancozeb on the mouse oocyte ultrastructure, evidencing overall proper preservation of oocytes exposed from 0.001 to 0.1 µg/mL. Morphology changes, mainly affecting organelle shape, density, and plasma membrane, were observed in oocytes exposed with the highest tested fungicide concentration (1 µg/mL). 

Mancozeb caused a reduction of thyroxine (T4) levels in female rats, neural tube defects [50,51], and genotoxic effects in humans [31]. The reproductive toxicity caused in vivo pathological alterations in mouse ovaries, impaired fertilization, and alterations of the estrous cycle [52,53]. Its action was responsible for a decrease in the number of healthy follicles and an increase in the number of atretic follicles [40,54]. These data were confirmed in vitro, where increasing doses of mancozeb (0.001–1 µg/mL) determined a reduction in the fertilization rates and an alteration of the oocyte meiotic spindle morphology [39]. 

At the lowest concentrations of mancozeb (0.001–0.01 μg/mL), our morphological data indicated good overall preservation of the oocyte ultrastructure; no specific alterations were detected on mitochondria, cortical granules, and microvilli. In the ooplasm of all groups examined, the presence of the “cytoplasmic lattice” formed by a fibrillar protein matrix was abundant [55,56]. These fibrillar structures could play a regulatory role in oocyte maturation by acting as a storage site for ribosomes and maternal ribosomal RNA during the early stages of embryo development [57,58]. Recently, the cytoplasmic lattice was associated with the subcortical maternal complex (SCMC), a multiprotein complex located in the cellular subcortex and inherited from the mother. The interplay between cytoplasmic lattice and SCMC seems to be critical for the oocyte-embryo transition, particularly for meiotic spindle formation and positioning, translation regulation, organelle redistribution, and epigenetic reprogramming [58]. 

However, ultrastructural changes were found in oocytes treated with 0.1 and 1 µg/mL of mancozeb. Our morphological findings showed the rearrangement, as well as the reduction of cell organelles/ratio in the ooplasm, related to the exposure dose. This non-homogeneous distribution pattern could be a sign of oocyte immaturity or be associated with loss of oocyte viability, at least in mice, as shown by previous studies [59,60,61]. TEM analysis revealed a reduction in the numerical density of mitochondria in mouse oocytes treated with mancozeb 1 µg/mL compared to controls. Moreover, in this experimental group, mitochondria appeared mostly vacuolated. Vacuolated mitochondria are a peculiar aspect in mouse oocytes, where they can present one or two light vacuoles in their matrix [62], but their role remains unclear. Some authors hypothesized they may represent an immature form limiting ROS [63,64]. Others believe that this specific mitochondrial morphology could be associated with an increased surface area due to the expansion of outer and inner membranes [65]. Vacuolated mitochondria could also result in a reduction of their membrane potential and a decreased efficiency [66]. Such ultrastructural changes have been reported in various pathological states and could indicate dysfunctions of these organelles and/or activation of apoptotic phenomena [67]. 

In line with this, the decrease in the numerical density of mitochondria at the highest concentration and a higher prevalence of their vacuolated morphology may indicate alterations in respiratory activity and energetic metabolism or oxidative stress conditions. In addition, mitochondria are maternally inherited, so genetic, functional, structural, and numerical abnormalities in the oocyte, associated with metabolic defects, could compromise the embryos’ ability to pass the pre-implantation stages. Mitochondrial defects, such as impaired membrane potential, altered mitochondrial DNA expression, and structural abnormalities, could lead to irrecoverable failure of the pre-implantation embryo [68]. 

Ultrastructure analysis revealed a significant reduction in the linear density of cortical granules in the group at 1 µg/mL, compared to control and mancozeb 0.001 µg/mL groups. Usually, cortical granules are small organelles, regularly present, stratified in one to three rows in the cortex region of the oocytes. They represent a product of the Golgi complex; they showed different electron densities in mice (dark or light), as observed by Nicosia and collaborators (1977) [69], which might depend on different stages of maturation, and they have a unique role in fertilization [70,71,72,73]. Indeed, cortical granules are usually involved in the blocking mechanism of polyspermia through the exocytosis of their contents (glycosaminoglycans, proteases, acid phosphatases, and peroxidases) in the perivitelline space, called “cortical reaction” [69,74,75]. The reduction and/or absence of these organelles, as suggested by our findings, could be a marker of premature exocytosis of its content and could be indicative of an inability of the oocyte to interact with the spermatozoa, leading to non-monospermic fertilization and the production of an embryo, with an inappropriate chromosome arrangement [75,76].

Furthermore, the results of our work reported an irregular and decreased distribution of microvilli, highly significant at the 1 µg/mL concentrations, which are short and sometimes flattened, compared with controls and other tested groups. Microvilli are dynamic structures in the oolemma, known for their role in membrane fusion during fertilization [77,78]. Some authors suggest that microvilli may act as a platform, which concentrates adhesion/fusion proteins and/or provides membranous protrusion with a slight curvature radius, facilitating the interaction between spermatozoa and oocytes [79,80]. Thus, alterations in microvilli morphology, emerging from our observations, could represent the inability of the oocyte to facilitate sperm entrance, reducing the fertilization rate. 

In all groups studied, it was detected the presence of multivesicular bodies, which characterized the cytoplasm of mouse oocytes. These structures are large pale spherical vacuoles containing rounded vesicles and represent a variety of lysosomes. Multivesicular bodies behave as autophagic vacuoles or autolysosomes and digest endogenous material such as secretory granules, thus regulating secretory processes within certain cells [81]. However, these structures, dense lamellar bodies, and phagophore-like structures at the different stages of maturation may represent autophagic phenomena, which could be associated with cellular stress conditions, as previously reported [76,82]. Moreover, the extracellular materials, exosome vesicles, and debris noted in the perivitelline space could be compatible with different processes, such as exocytosis of residual bodies or autophagic exocytosis. These data represent evidence of degenerative changes related to the apoptotic process or aging of the oocyte [83,84] due to the accumulation of autophagic or degradative vesicles in the cytoplasm. 

Our previous in vitro study demonstrated dose-related toxicity of increasing concentrations of mancozeb (from 0.001 to 1 µg/mL) on the ultrastructure mouse granulosa cells, resulting in intercellular contact alterations, nuclear membrane irregularities, chromatin marginalization and condensation, membrane blebbing and signs of apoptosis [42,85]. Recently, it was found that mancozeb suppresses granulosa cells’ viability and changes their morphology but induces granulosa cells to secrete progesterone, which could inhibit LH (luteinizing hormone) production and suppress ovulation [37]. Since granulosa cells contribute to the development and maturation of oocytes, their alterations may be associated with the oocyte’s reduced capability and sterility [86,87,88]. However, in this current study, no dose-dependent toxicity of mancozeb on mouse oocyte ultrastructure was found. Emerged data from our work showed some ultrastructural changes at the highest concentration, which could be attributed to the fungicide, but without indicating gradual alterations depending on the exposure dose. This could be connected to the protective role against toxicants exerted by cumulus cells toward the oocyte, as previously demonstrated by others [89,90]. In our experimental model, cumulus cells exerted quite efficient protection at low concentrations of mancozeb, becoming ineffective at 1 µg/mL. Our morphological findings added information regarding the potentially harmful impact of mancozeb on the mammalian oocyte ultrastructure.

## 5. Conclusions

In conclusion, we reported ultrastructural changes on the mouse oocyte primarily at the highest concentration of mancozeb, indicating the morphological location of the alterations and the individual compartments altered and explaining its detrimental effects on female reproductive health. The interesting aspect of this study is that mancozeb-induced detrimental effects on the ultrastructure of mouse oocytes only at the highest concentration (1 µg/mL) could be indicative of protection exerted by nourishing cumulus cells toward the oocyte. However, this action seems to be not fully effective at the highest concentration, being responsible for the toxicant intake by the oocyte. These results are of interest for fertility preservation and infertility studies, also because using oocytes matured in vitro provides an easy and useful experimental model of reproductive toxicity to study the harmful effects of pesticides, with results potentially transferable to higher species, including humans.

## Figures and Tables

**Figure 1 biology-12-00698-f001:**
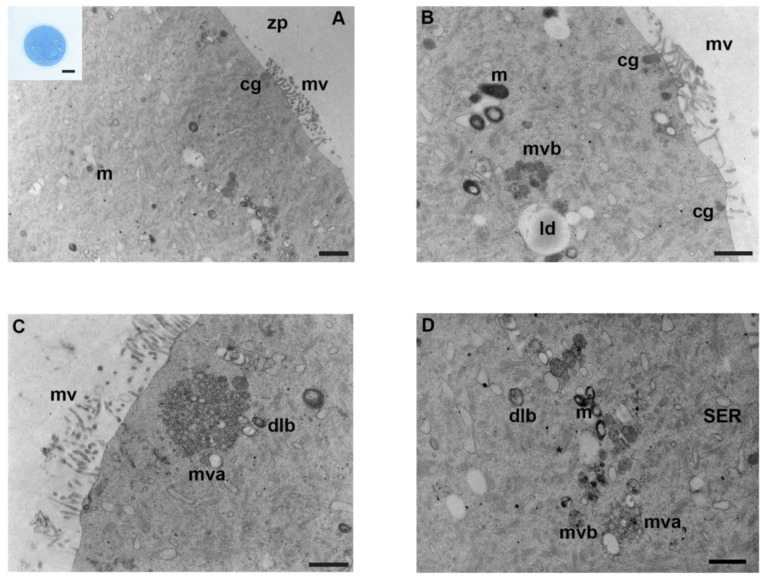
Ultrastructure of mouse oocytes in the controls group. Representative TEM micrographs showing in (**A**) the general morphology of the cortical region in MII mouse oocytes, microtopography of intracellular organelles, and microvillar processes. Round/ovoid mitochondria (m) and cortical granules (cg) are visible; zp: zona pellucida; mv: microvilli (TEM, bar: 1 µm). Inset in (**A**): a representative image of a semithin section of mouse oocyte (LM, Mag: 40×). (**B**) High magnification, cortex of mouse oocytes evidence clusters of mitochondria (m), lipid droplets (ld), multivesicular bodies (mvb), cortical granules (cg), and regular distribution of microvilli (mv) on the oolemma (TEM, bar: 800 nm). (**C**) Multivesicular aggregates (mva) are visible in the cortex, with dense lamellar bodies (dlb), at high magnification. Notice long and thin microvilli (mv) (TEM, bar: 800 nm). (**D**) Portion of ooplasm showing cell organelles: mitochondria (m) with electron-dense cristae, accompanied by multivesicular bodies (mvb), multivesicular aggregates (mva). Dense lamellar bodies (dlb), SER and the extensive fibrillar matrix of cytoplasmic lattice (*) are observed (TEM, bar: 600 nm).

**Figure 2 biology-12-00698-f002:**
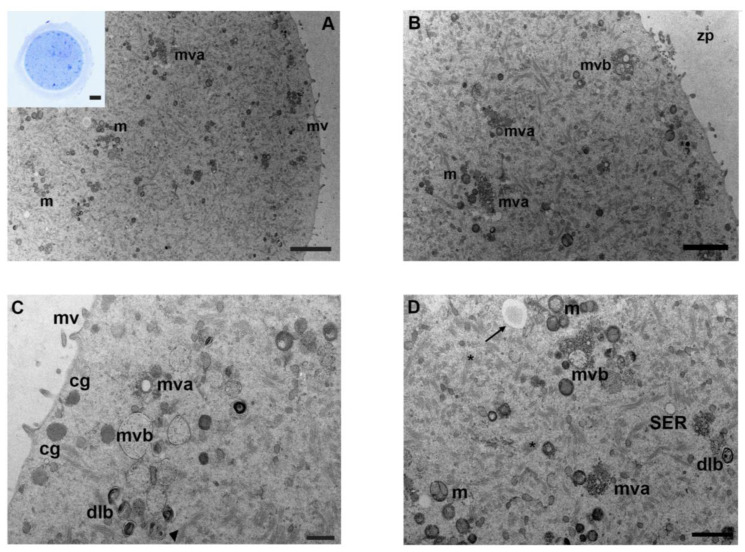
Ultrastructure of mouse oocytes in mancozeb 0.001 µg/mL group. (**A**) Low magnification of TEM micrographs from MII mouse oocytes showing high preservation of cell organelles, homogeneously distributed in the cytoplasm. Clustered mitochondria (m) and numerous multivesicular aggregates (mva) are visible; mv: microvilli.(TEM, bar: 2 µm). Inset in (**A**): a representative semithin section of mouse oocyte (LM, Mag: 40×). (**B**) Representative TEM image of the cortical region in MII mouse oocytes. Clusters of mitochondria (m), with electron-dense cristae and matrix, are visible, accompanied by multivesicular aggregates (mva) and multivesicular bodies (mvb). zp: zona pellucida (TEM, bar: 500 nm). (**C**) At high magnification, a small portion of the cortex evidences multivesicular bodies (mvb), with dense lamellar bodies (dlb) and multivesicular aggregates (mva). Cortical granules (cg) are linearly arranged below the oolemma. Note multivesicular bodies (mvb) and dense lamellar bodies (dlb) are in close association with an autophagic-like vesicle (arrowhead). Short and thick microvilli (mv) are observed (TEM, bar: 500 nm). (**D**) Micrographs of cytoplasmic ultrastructure in mouse oocyte. Image shows different cytoplasmic structures as mitochondria (m), with a round or oval shape and visible double membranes, multivesicular bodies (mvb) and aggregates (mva), dense lamellar bodies (dlb), and fibrillar matrix of cytoplasmic lattice (*). Immature autophagic-like vesicle delimited by a double membrane and a wider lumen (arrow). SER, with small vesicles, is also visible (TEM, bar: 1 µm).

**Figure 3 biology-12-00698-f003:**
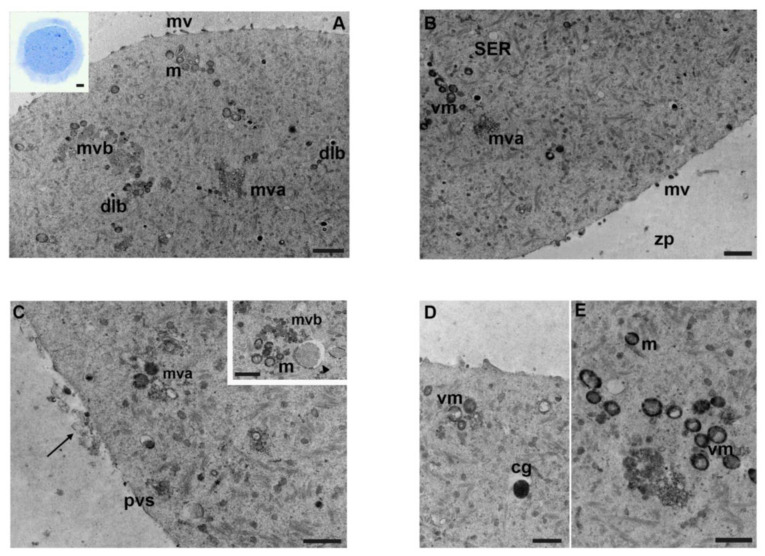
Ultrastructure of mouse oocytes in mancozeb 0.01 µg/mL group. Representative TEM micrograph of mouse oocyte showing in (**A**) patchy distribution of intracellular organelles. Note numerous dense lamellar bodies (dlb), multivesicular bodies (mvb), multivesicular aggregates (mva), and mitochondria (m); mv: microvilli (TEM, bar: 1 µm). Inset in (**A**): representative semithin section of mouse oocyte (LM, Mag: 40×). (**B**) A portion of free organelles in the cortical region. Clusters of vacuolated mitochondria (vm), multivesicular aggregates (mva), andSER are observed. Cortical granules are less visible. Less, short, and thicker microvilli (mv) are present. zp: zona pellucida (TEM, bar: 1 µm). (**C**) High magnification of cortical region in MII oocytes showing few organelles. Notice, in the perivitelline space (pvs), extracellular material and debris (arrow); mva: multivesicular aggregates; (*): cytoplasmic lattice (TEM, bar: 800 nm). Inset in (**C**): clusters of mitochondria (m), multivesicular bodies (mvb), and immature autophagic-vesicle (arrowhead) delimited by a double membrane, with recognizable material derived from cytoplasmic organelles (TEM, bar: 800 nm). (**D**) Isolated cortical granule (cg) visible below the oolemma; vacuolated mitochondria (vm) (TEM, bar: 800 nm). (**E**) Groups of vacuolated mitochondria (vm) and mitochondria (m) at high magnification (TEM, bar: 800 nm).

**Figure 4 biology-12-00698-f004:**
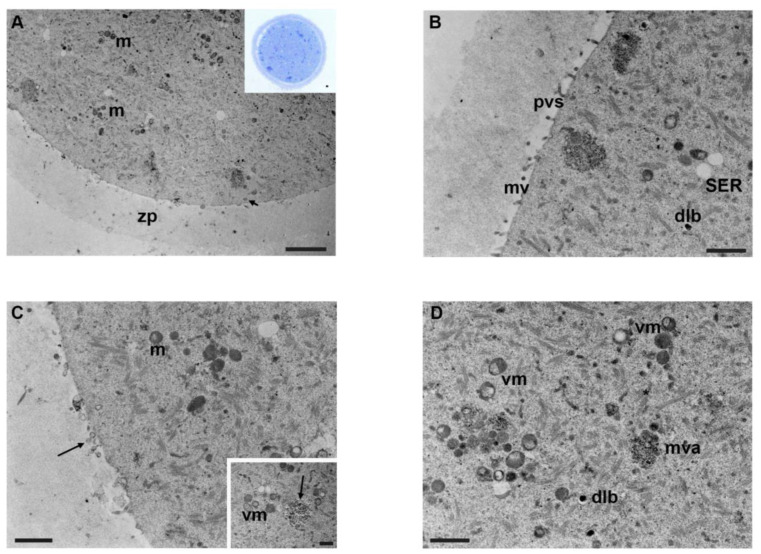
Ultrastructure of mouse oocytes in mancozeb 0.1 µg/mL group. (**A**) TEM micrograph showing general morphology of MII oocyte. Few organelles are visible in the cytoplasm; m: mitochondria; zp: zona pellucida. Rare cortical granules are visible (arrow) (TEM, bar: 2 µm). Inset in (**A**): semithin section of mouse oocytes (LM, Mag: 40×). (**B**) A portion of the cortical region showing nonhomogeneous and short microvilli (mv) protruded in the perivitelline space (pvs), SER, and dense lamellar bodies (dlb) (TEM, bar: 1 µm). (**C**) Cortex of mouse oocytes with few organelles, sporadic clusters of mitochondria (m). Notice the presence of extracellular materials and debris (arrow) in the perivitelline space. (TEM, bar: 1 µm). Inset in (**C**): Mature autophagic-like vesicles (arrow) enclosed in a single membrane, containing material of unrecognizable origin and vacuolated mitochondria (vm) (TEM, bar 600 nm). (**D**) TEM image of ooplasm showing vacuolated mitochondria (vm), dense lamellar bodies (dlb), multivesicular aggregates (mva), and abundant cytoplasmic lattice (*) (TEM, bar: 1 µm).

**Figure 5 biology-12-00698-f005:**
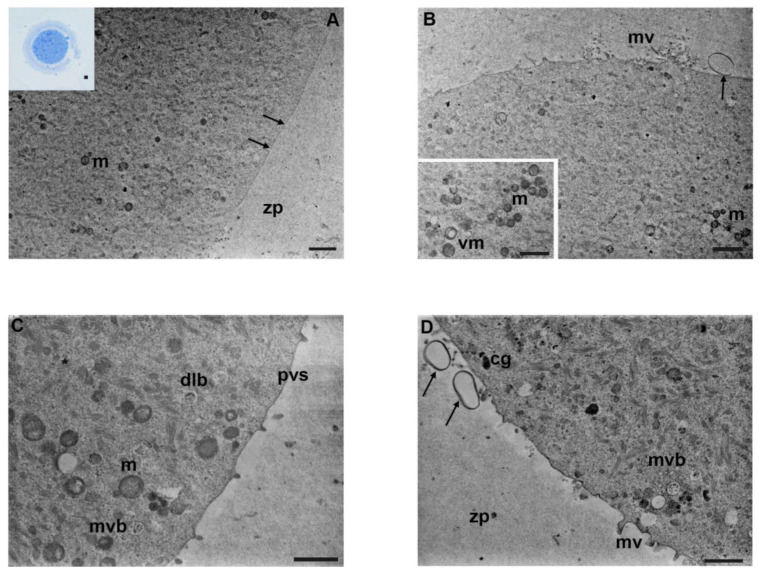
Ultrastructure of mouse oocytes in mancozeb 1 µg/mL group. (**A**) Representative TEM micrograph of the cortical region showing low cell organelles density. Few mitochondria (m) are visible; cortical granules are absent. Notice the lack of microvilli (arrow) on the oolemma; zp: zona pellucida (TEM, bar: 2 µm). Inset in (**A**): semithin section of mouse oocytes, with condensed organelles in one pole (LM, Mag: 40×). (**B**) A portion of the cortical region with a non-homogeneous distribution of organelles beneath the oolemma and short and thick microvilli (mv). Exocytotic vesicles (arrow) are visible in the perivitelline space; mitochondria (m) (TEM, bar: 1 µm). Inset in (**B**): groups of mitochondria (m) and vacuolated mitochondria (vm) (TEM, bar: 1 µm). (**C**) High magnification of cortex part with more organelles present; mitochondria (m) with evident cristae, dense lamellar bodies (dlb), and multivesicular bodies (mvb); pvs: perivitelline space; (*): cytoplasmic lattice; (TEM, bar: 1 µm). (**D**) Representative TEM micrographs of the cortical region in mouse oocyte. Multivesicular bodies (mvb) are visible. Few cortical granules (cg) are present. Microvilli (mv) are short, and irregularly distributed on the oolemma. Extracellular material, exosomes, and debris (arrow) are detected in the perivitelline space; zp: zona pellucida; (TEM, bar: 1 µm).

**Table 1 biology-12-00698-t001:** Summary of the qualitative data obtained by TEM analysis on the main ultrastructural features in oocytes unexposed (controls) or exposed to increasing concentrations of mancozeb (0.001–1 µg/mL).

			Mancozeb		
	Control	0.001 µg/mL	0.01 µg/mL	0.1 µg/mL	1 µg/mL
**Cytoplasmic lattice**	Uniformly distributed	Uniformly distributed	Uniformly distributed	Uniformly distributed	Uniformly distributed
**Mitochondria**	Round to ovoid shaped, with dense matrix	Round to ovoid shaped	Increased vacuolated forms	Increased vacuolated forms	Prevalent vacuolated forms
**Cortical Granules**	Round, dark electron density	Round, dark electron density	Round, dark electron density	Round, dark electron density	Round, dark electron density
**Microvilli**	Long, and thin	Thicker	Short and thick	Short and thick	Flattened
**Zona pellucida**	Dense	Dense	Dense	Thin and dense	Thin and dense

**Table 2 biology-12-00698-t002:** Morphometric evaluation (expressed as mean ± standard deviation) of organelles in control and mancozeb-exposed groups (0.001–1 µg/mL). Morphometry was performed using one-way ANOVA with Tukey’s HSD post-hoc analysis. Different superscripts indicate a significant difference (*p* < 0.05).

			Mancozeb		
	Control	0.001 µg/mL	0.01 µg/mL	0.1 µg/mL	1 µg/mL
**Mitochondria (50 µm^2^)**	28 ± 7.131 ^a^	24.13 ± 5.436 ^a,b^	23.38 ± 8.684 ^a,b^	21.13 ± 4.794 ^a,b^	18.5 ± 5.175 ^b^
**Multivesicular bodies and dense lamellar bodies (50 µm^2^)**	7 ± 1.581	8.6 ± 1.517	7.4 ± 1.517	7.2 ± 0.836	7.4 ± 1.14
**Autophagic vesicles (50 µm^2^)**	0.6 ± 0.547	0.8 ± 0.836	1.4 ± 0.547	1.6 ± 0.547	1.8 ± 0.836
**SER (50 µm^2^)**	3 ± 1	2.8 ± 1.789	2.4 ± 1.14	1.8 ± 0.836	0.8 ± 0.836
**Cortical granules/10 µm**	3 ±0.707 ^a^	3.2 ± 0.837 ^a,b^	1.6 ± 0.894 ^a,c^	1.6 ± 1.14 ^a,c^	0.8 ± 0.837 ^c^
**Microvilli/10 µm**	15.6 ± 2.408 ^a^	14.4 ± 2.702 ^a,b^	11 ± 2.345 ^b^	11.4 ± 1.517 ^a,b^	6 ± 2.449 ^c^

## Data Availability

The data presented in this study are available on request from the corresponding authors.

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
