# Peer review of "Ultrastructural Evaluation of Mouse Oocytes Exposed In Vitro to Different Concentrations of the Fungicide Mancozeb"

_biology, 2023, doi:10.3390/biology12050698_

Round 1

Reviewer 1 Report

Summary

Author evaluated the effects on the ultrastructure of mouse oocytes isolated from cumulus-oocyte complexes (COCs), exposed in vitro to increasing concentrations of mancozeb. In summary, ultrastructural data including mitochondria, cortical granule and microvilli revealed changes mainly at the highest concentration of mancozeb on mouse oocytes. This could be responsible for the previously described impaired capability in oocyte maturation, fertilization, and embryo implantation, demonstrating its impact on reproductive health and fertility.

Major comment

In a previous report, you reported that manczeb affects granulosa cells. Is the adverse effect on the oocyte this study through the cumulus cells, or is it directly affecting the oocyte?

Minor comment

Table 1: Please evaluate numerically.

Table 2: Although there is no significant difference at 13 of 0.001ug, there is a significant difference at 18.5 of 1ug. Please reconsider the statistical analysis. Alternatively, the 1ug value may be wrong.

Author Response

Reviewer 1

Author evaluated the effects on the ultrastructure of mouse oocytes isolated from cumulus-oocyte complexes (COCs), exposed in vitro to increasing concentrations of mancozeb. In summary, ultrastructural data including mitochondria, cortical granule and microvilli revealed changes mainly at the highest concentration of mancozeb on mouse oocytes. This could be responsible for the previously described impaired capability in oocyte maturation, fertilization, and embryo implantation, demonstrating its impact on reproductive health and fertility.

Major comment

In a previous report, you reported that mancozeb affects granulosa cells. Is the adverse effect on the oocyte this study through the cumulus cells, or is it directly affecting the oocyte?

ANSWER: Thank you for the interesting question. We assumed that mancozeb-induced damage to the oocytes is mediated by cumulus cells, as demonstrated for other toxicants (please see ref. n 90 and 91). Results from the present study evidenced that CCs are responsible for effective protection toward the oocytes at low mancozeb doses. On the contrary, the dose of 1 μg/ml induces ultrastructural damage to the oocytes, thus demonstrating the failure of CCs in avoiding the transfer of mancozeb to the oocyte, via transzonal projections. According to your suggestion, we added a short paragraph to better explain this concept (please, see lines 538-541 in the final version).

Minor comment

Table 1: Please evaluate numerically.

ANSWER: Ok, done. We reported qualitative and quantitative data in two different tables to better explain the results obtained by TEM. Values related to the structure called “cytoplasmic lattice” were not reported due to its fibrillar structure (typical of the ooplasm and homogeneously distributed in all groups studied), not properly quantifiable.

Table 2: Although there is no significant difference at 13 of 0.001ug, there is a significant difference at 18.5 of 1ug. Please reconsider the statistical analysis. Alternatively, the 1ug value may be wrong.

ANSWER: Thank you, we noticed a typing error in the table. The wrong reported value of “13” was now correctly modified to “24.13”. Following your previous suggestion about Table 1, we decided to do one table that grouped all morphometric values.  

Reviewer 2 Report

In the present study, the authors found that mancozeb exposure affect the mitochondria, cortical granules and microvilli structure by using light and transmission electron microscopy.  Although the toxic evidences of mancozeb exposure ((0.001-1 μg/ml) to organelle structure is obvious, but what is the concentration of environmental exposure? Is that convinced by performing in in vitro experiment? Therefore, we would not get credible conclusion until the environmental exposure model have been set up and the general toxic effects of chemical compounds have been excluded.

Author Response

Reviewer 2

In the present study, the authors found that mancozeb exposure affect the mitochondria, cortical granules and microvilli structure by using light and transmission electron microscopy.  Although the toxic evidences of mancozeb exposure (0.001-1 μg/ml) to organelle structure is obvious, but what is the concentration of environmental exposure? Is that convinced by performing in in vitro experiment? Therefore, we would not get credible conclusion until the environmental exposure model have been set up and the general toxic effects of chemical compounds have been excluded.

ANSWER: Thank you for your questions and comments. In responding regarding the concentration of environmental exposure, it is important to note that mancozeb has been used in agricultural settings for decades, which suggests that its concentration may vary depending on factors such as climate, soil type, and crop type.

According to your comments, we have reviewed available data about the environmental contamination of mancozeb in the introduction (please, see lines 61-87 in the final version). Briefly, the US. EPA estimated the environmental surface water concentration of mancozeb to be 0.1 to 25.2 µg/L (peak exposure) and 0.1 µg/L for long-term exposure.  Kollman in the report entitled “Summary of assembly bill 1807/3219, pesticide air monitoring results” (1995) estimated the air monitoring value of mancozeb in California (probably associated with other particles) to be between 0.29 µg/m3 (0.02 ppb) and 1.81 µg/m3 (0.13 ppb).

Mancozeb was classified as a fungicide with low acute toxicity and low persistence in the environment (the reason why it has been used and continues to be used in many countries). Therefore, the main cause of adverse effects seems to be chronic exposure at low doses. In vitro experiments here performed allowed us to study the effects of mancozeb exposure on organelle ultrastructure, which can provide an understanding of how varying concentrations of the chemical may affect germ cells. The choice of concentrations was based on previous in vitro studies, in which oocyte alterations were demonstrated (see lines 83-87 and 106-127 in the final version).

Round 2

Reviewer 2 Report

The manuscript is available to be accepted now.